# The Extract of *Arctium lappa* L. Fruit (Arctii Fructus) Improves Cancer-Induced Cachexia by Inhibiting Weight Loss of Skeletal Muscle and Adipose Tissue

**DOI:** 10.3390/nu12103195

**Published:** 2020-10-19

**Authors:** Yo-Han Han, Jeong-Geon Mun, Hee Dong Jeon, Dae Hwan Yoon, Byung-Min Choi, Ji-Ye Kee, Seung-Heon Hong

**Affiliations:** 1Department of Oriental Pharmacy, College of Pharmacy, Wonkwang-Oriental Medicines Research Institute, Wonkwang University, Iksan, Jeonbuk 54538, Korea; dygks1867@hanmail.net (Y.-H.H.); wjdrjs92@daum.net (J.-G.M.); alen0707@naver.com (H.D.J.); ydh2715@gmail.com (D.H.Y.); 2Department of Clinical and Administrative Pharmacy, College of Pharmacy, University of Georgia, Augusta, GA 30901, USA; 3Department of Biochemistry, School of Medicine, Wonkwang University, Iksan, Jeonbuk 54538, Korea; bmchoi@wonkwang.ac.kr

**Keywords:** Arctii Fructus, cancer cachexia, browning, 3T3-L1 cells, C2C12 myoblast

## Abstract

Background: Cachexia induced by cancer is a systemic wasting syndrome and it accompanies continuous body weight loss with the exhaustion of skeletal muscle and adipose tissue. Cancer cachexia is not only a problem in itself, but it also reduces the effectiveness of treatments and deteriorates quality of life. However, effective treatments have not been found yet. Although Arctii Fructus (AF) has been studied about several pharmacological effects, there were no reports on its use in cancer cachexia. Methods: To induce cancer cachexia in mice, we inoculated CT-26 cells to BALB/c mice through subcutaneous injection and intraperitoneal injection. To mimic cancer cachexia in vitro, we used conditioned media (CM), which was CT-26 colon cancer cells cultured medium. Results: In in vivo experiments, AF suppressed expression of interleukin (IL)-6 and atrophy of skeletal muscle and adipose tissue. As a result, the administration of AF decreased mortality by preventing weight loss. In adipose tissue, AF decreased expression of uncoupling protein 1 (UCP1) by restoring AMP-activated protein kinase (AMPK) activation. In in vitro model, CM increased muscle degradation factors and decreased adipocytes differentiation factors. However, these tendencies were ameliorated by AF treatment in C2C12 myoblasts and 3T3-L1 cells. Conclusion: Taken together, our study demonstrated that AF could be a therapeutic supplement for patients suffering from cancer cachexia.

## 1. Introduction

Cancer-induced cachexia is a wasting syndrome related to various cancers, affecting body weight with loss of skeletal muscle and adipose tissue [1]. Because of these symptoms, patients who suffer from cachexia experience anorexia, fatigue, anemia, and edema [2]. These symptoms reduce cancer patients’ quality of life and reduce the efficiency of therapies such as chemotherapy and radiotherapy [3]. Furthermore, systemic inflammatory responses and skeletal muscle loss are caused by cancer, and chemotherapy makes the cachexia much worse [4,5,6]. More than half of cancer patients suffer from cachexia, and approximately 20% of them die because of cachexia [7]. However, there is still no apparent treatment drug.

Skeletal muscle atrophy is one of the major features of cancer-induced cachexia. Previous studies have shown that skeletal muscle is degraded by the E3 ubiquitin ligase, including muscle atrophy F-box (MAFbx) and muscle-specific RING finger protein (MuRF-1) [8]. Since MAFbx and MuRF-1 are significantly increased in the skeletal muscle of cancer patients, they are considered as markers of muscle atrophy [9,10]. Interleukin (IL)-6, which is increased by viral infection and inflammation in cancer, exhibited a pleiotropic activity. Although the low concentration of IL-6 can activate and regenerate myotube, chronic exposure to IL-6 induces the degradation of muscle tissues [11].

In advanced cancer patients, the loss of adipose tissue increases mortality. Some studies have recently reported that cancer patients with low and normal BMI have shown higher mortality than obese patients [12]. It is also reported that systemic inflammation induced by cancer has shown to not only promote lipolysis of adipose tissue but also increase browning [13]. Peroxisome proliferator-activated receptor γ (PPARγ) and CCAAT/enhancer-binding protein α (C/EBPα) are transcriptional factors and induce most of the gene expression in adipogenesis [14]. However, these genes are substantially reduced in the early stages of cachexia, and wasting of adipose tissue is followed by impaired lipid metabolism, which is an imbalance between lipolysis and lipogenesis [15]. For this reason, despite sufficient nutrition, the energy consumption of cancer patients is increased, and eventually their survival period is shortened [16]. In addition, browning of white adipose tissue (WAT) is associated with cancer-induced cachexia [13,17]. Browning adipocytes, which express uncoupling protein 1 (UCP1), increase energy expenditure, and eventually reduce fat mass [18].

In this study, we used Arctii Fructus (AF), the dried fruit of *Arctium lappa* used as herbal medicine in East Asia [19]. AF improves energy homeostasis by regulating adipocyte differentiation in vitro and in vivo [20]. Moreover, AF inhibits colorectal cancer cell growth and its metastasis to the lung [19]. AF exerts anti-inflammatory effects in human mast cell-mediated allergic responses [21]. Since cachexia is a metabolic disorder caused by systemic inflammation in cancer patients, it is expected that AF can improve cachexia by regulating inflammation and energy balance. However, there is no report on the therapeutic effect of AF on cancer-induced cachexia. Therefore, this study investigated the improvement effect of AF on cancer-induced cachexia in vivo and in vitro.

## 2. Materials and Methods

### 2.1. Reagents and Antibodies

5-aminoimidazole-4-carboxamide-1-β-d-ribofuranoside (AICAR) was purchased from Med Chem express (Monmouth Junction, NJ, USA). Oil Red O solution, DMSO, isopropanol, insulin, 3-isobutyl-1-methylxanthine (IBMX), dexamethasone (Dex), arctiin and arctigenin were obtained from Sigma-Aldrich (St. Louis, MO, USA). Bovine serum (BS), fetal bovine serum (FBS), horse serum (HS), Dulbecco’s Modified Eagle’s Medium (DMEM), penicillin-streptomycin were purchased from Gibco BRL (Grand Island, NY, USA). Mouse IL-6 enzyme-linked immunosorbent assay (ELISA) kit was purchased from BD Biosciences (Mountain View, CA, USA). Antibodies against MAFbx, MuRF1, and Glyceraldehyde 3-phosphate dehydrogenase (GAPDH) were purchased from Santa Cruz Biotechnology (Santa Cruz, CA, USA). PPARγ, C/EBPα, and phospho AMP-activated protein kinase (AMPK) antibodies were purchased from Cell Signaling (Danvers, MA, USA). UCP1 and myosin heavy chain (MyH) antibody was purchased from Invitrogen (Carlsbad, CA, USA).

### 2.2. Extract of AF

Dried AF was purchased from Omniherb (Uiseong, Korea). Water extract of AF (WAF) and ethanol extract of AF (EtAF) were prepared as we previously described [20]. Briefly, AF was extracted with water (100 g/L of water) and 70% ethanol (100 g/L) for 3 h. The yields of WAF and EtAF were 6.8% and 13.7%, respectively.

### 2.3. HPLC Analysis of AF

Chromatographic analysis was performed on Younglin HPLC YL9100 system (Anyang, Korea) consisting of a YL9110 quaternary gradient pump, YL9160 PDA Detector. All samples were separated on a SHISEIDO PAK C18 column (250 mm × 4.6 mm, 5 μm). The mobile phase consisting of acetonitrile (A) and water (B) was run at 0.7 mL/min. The solvent system was set as follows: 0–20 min, 20% A→30% A; 20–35 min, 30% A→36% A; 35–50 min, 36% A→45% A; and 50–52 min, 45% A→90% A, respectively. The detection wavelength was 280 nm and the column temperature was 30 °C.

### 2.4. Cachexia Mice Model by Colorectal Cancer Cells Injection

Male BALB/c mice (8 weeks old) were purchased from Samtaco Science (Osan, Korea). Before the injection of CT-26 cells, mice were adapted in an experimental facility (22 ± 2 °C under a 12 h light/dark cycle). To induce a mildly cachectic mice model, we subcutaneously inoculated the mice with CT-26 cells (1 × 10^6^ cells per mouse) in the abdominal area. On day 9 after CT-26 cells inoculation, we randomly divided the mice into five groups (*n* = 5) and administrated WAF, EtAF, and AICAR until the end of the experiment. WAF and EtAF were orally administered at a dose of 100 mg/kg every day. In our experiment, we used AICAR as a positive control. AICAR, which is an AMPK activator, has been reported as able to recover muscle wasting induced by inflammation [22]. AICAR was intraperitoneally injected at 250 mg/kg every 2 days. The control group and tumor group mice were administered with the same volume of water. The protocol for mildly induced cachectic mice was shown in Appendix A.

To induce a severely cachectic mice model, we injected CT-26 cells (1 × 10^6^ cells per mouse) intraperitoneally using a 1 mL syringe to mimic diffuse carcinomatosis. On day 7 after CT-26 cells injection, we randomly divided the mice into four groups (*n* = 10). WAF and EtAF at a dose of 100 mg/kg were orally administered once a day until the end of the experiment. The protocol for severely induced cachectic mice was shown in Appendix A.

Tumor volume was determined by the electric caliper and calculated by the modified ellipsoid formula 1/2 (Length × Width^2^). For each cage, food intake was measured and then divided by the number of mice. For the serum analysis, the mice fasted for 3 h before sacrifice. The serum was separated by centrifugation at 4000× *g* for 30 min. The aspartate aminotransferase (AST), alanine aminotransferase (ALT), blood urea nitrogen (BUN), and creatinine were assayed by the Seoul Medical Science Institute (Seoul Clinical Laboratories, Seoul, Korea). Animal experiments conducted in this study were approved by the Institutional Review Board of Wonkwang University (WKU20-40).

### 2.5. Hematoxylin and Eosin (H&E) Staining

All resected gastrocnemius muscle tissues (gMuscle) and epididymal white adipose tissue (eWAT) were fixed in 10% neutral buffered formalin for 24 h. After paraffin embedding, embedded tissues were sliced to 5 μm. The sections were stained with H&E. Microscopic images of stained tissues were taken by the EVOS XL Core Imaging System (Thermo Fisher Scientific, Waltham, MA, USA). The magnification was ×200, and the scale bar was 100 µm. The average size of adipocyte was analyzed by ImageJ software (National Institutes of Health, Bethesda, MD, USA). Briefly, adipocytes size was analyzed using 10 randomly selected areas of each H&E-stained WAT image. Adipocyte number was counted by randomly selecting fields at 200× magnification of each image.

### 2.6. CT-26 Conditioned Media (CM)

To mimic the cancer status administered with WAF and EtAF, CT-26 cells cultured media treated with WAF and EtAF were used in vitro as CM for 3T3-L1 cells and C2C12 cells. We prepared CM from CT-26 cells according to the previous report with slight modifications [23]. Briefly, CT-26 cells were seeded in 100 mm cell culture plate at 5 × 10^6^ cells and maintained for 24 h in 10% FBS/DMEM media containing 1% penicillin/streptomycin. Then, CT-26 cells were treated with WAF and EtAF at a dose of 50 and 100 µg/mL for 24 h. After washing two times using PBS, cells were incubated for 48 h in 6 mL of 1% FBS/DMEM media. WAF-CM, CM treated with WAF; EtAF-CM, CM treated with EtAF. Debris were removed from the collected media and diluted for C2C12 and 3T3-L1 cells culture.

### 2.7. Measurement of IL-6

To measure IL-6 in mice serum and cell cultured media, we performed ELISA according to the manufacturer’s instructions. Briefly, to confirm the IL-6 production in serum, serum was collected in a serum separator tube and then centrifuge for 20 min at 1000× *g*. For the CT-26 cells culture media, we used CM for determining the IL-6 levels.

### 2.8. C2C12 Cell Differentiation

C2C12 cells were purchased from American Type Culture Collection (ATCC) (Rockwill, MD, USA). C2C12 cells were grown in DMEM containing 10% FBS, 1% penicillin-streptomycin. To inducing myogenic differentiation, C2C12 cells were seeded in a 6-well plate and cultured until 80–90% confluence status. The cultured media were changed into differentiation media consisting of 2% HS/DMEM containing 1% penicillin-streptomycin. To mimic cachectic symptom, C2C12 cells were cultured in the differentiation media consisting 50% CM for 36 h. The media were changed with fresh differentiation media every 2 days until day 6.

### 2.9. 3T3-L1 Cell Differentiation

3T3-L1 cells were purchased from ATCC. Using DMEM consisting of 1% penicillin-streptomycin and 10% BS, 3T3-L1 cells were cultured in at 37 °C in a 5% CO_2_ atmosphere. The 3T3-L1 pre-adipocytes were differentiated into mature adipocytes as we previously reported [24]. To mimic cachectic symptoms, 3T3-L1 cells were maintained in the differentiation media containing 10% CT-26 CM. From day 0 to day 1, 3T3-L1 cells were maintained in CM containing differentiation media, and the media were changed to fresh differentiation media for an additional 24 h. Then, until fully differentiated into mature adipocyte, we changed media to fresh media with insulin.

### 2.10. Cell Viability

To determine cell viability, we used MTS solution (Promega Corporation, Madison, WI, USA). Briefly, C2C12 and 3T3-L1 cells were seeded in 96-well plates at 3 × 10^3^ cells/well and the cells were incubated with CM, WAF, and EtAF. Then, 10 µL of MTS solution was treated for 4 h in an incubator. The absorbance was measured at 490 nm with a VERSA max microplate reader (Molecular Devices, Sunnyvale, CA, USA).

### 2.11. Oil Red O Staining

Lipid accumulation in 3T3-L1 cells was measured using Oil Red O solution. In brief, 3T3-L1 cells were fixed in 10% formaldehyde for 1 h. After washing with 60% isopropanol, cells were stained with 0.5% Oil Red O solution for 30 min. The cells were observed with a microscope and the images were captured by a microscope (Leica, Wetzlar, Germany). Following the microscopic observation, the stained solution was dissolved in 100% isopropanol, and absorbance was read using a VERSA max microplate reader at 500 nm.

### 2.12. Western Blot Analysis

Cells (5 × 10^5^ cells/well) were cultured in a 6-well plate and then lysed by lysis buffer (iNtRON Biotech, Seongnam, Korea). The cell lysates and 2× sample buffer were mixed and boiled at 95 °C for 5 min. The samples were resolved on SDS-page gels and were transferred onto Polyvinylidene fluoride or polyvinylidene difluoride membranes. The membranes were divided into several parts of each target proteins. To confirm the target proteins, we used primary and secondary antibodies, according to the manufacturer’s protocols. Images were taken using the FluorChem E system (ProteinSimple, San Jose, CA, USA). Detected bands were normalized by GAPDH using ImageJ software (National Institute of Health, Bethesda, MD, USA).

### 2.13. Quantitative Real-Time RT-PCR

Total RNA was collected from tissues and cells using a QIAzol lysis reagent (QIAGEN sciences, Germantown, MD, USA). First-strand cDNA was synthesized using 2000 ng of RNA and a Power cDNA Synthesis Kit (Applied Biosystems, Carlsbad, CA, USA). mRNA levels were confirmed using a Power SYBR^®^ Green PCR Master Mix and a StepOnePlus Real-Time PCR System (Applied Biosystems, Foster City, CA, USA). The primer sequences are listed in Table 1. Target genes were normalized by GAPDH and calculated using the ddCt method.

### 2.14. Statistical Analysis

The results are expressed as the mean ± SD of independent experiments. In vivo, we used Kruskal–Wallis test with Dunn’s multiple comparisons. In vitro, statistical evaluations were performed by a one-way ANOVA with Tukey’s multiple comparison test and two-way ANOVA using GraphPad PRISM software version 8 (GraphPad PRISM software Inc., San Diego, CA, USA). Values of *p* < 0.05 were considered to represent a significant difference.

## 3. Results

### 3.1. HPLC Analysis

Arctiin and arctigenin are the main bioactive compounds of AF [25]. Arctiin and arctigenin are natural compounds belonging to the lignin family and are characterized by their dibenzylbutane skeleton [26]. Lignans are plant compounds that have various bioactivities for preventing several diseases such as obesity, cancer and cardiovascular diseases [27,28]. Therefore, various pharmacological effects could be expected from plant compounds, including arctiin and arctigenin. It has been reported that arctigenin and arctiin account for approximately 0.5–2% (*w*/*w*) and 2–10% (*w*/*w*) of the dry weight of the AF, respectively [29]. These compounds also possessed an anti-inflammatory effect. Arctiin could inhibit pro-inflammatory mediators such as IL-1β, IL-6, TNF-α, and PGE2 through inactivation of NF-κB [30]. In addition, arctigenin showed an inhibitory effect on inflammation by reducing IL-6, TNF-α, NO, and COX-2 expression in activated macrophages [31]. In the previous study, we already proved that the inhibitory effect of AF on metastatic colorectal cancer and allergic responses was derived from arctigenin [32]. To confirm whether arctiin and arctigenin are contained in WAF and EtAF, we conducted HPLC analysis. Arctiin and actigenin were identified in WAF and EtAF. In our experiment, the effect of WAF was relatively better than that of EtAF. Actttin was detected in WAF approximately 10% more than EtAF, and arctigenin was detected in EtAF approximately 30% more than WAF. From the HPLC results, the cachexia-improving effect of AF was more closely related to arctiin than arctigenin (Figure 1).

### 3.2. AF Treatment in Mild Cachectic Mice Inhibited Weight Loss and Improved Cachectic Symptoms

To induce weight loss, cancer cells were injected subcutaneously into the mice. Since 10% of weight loss due to cancer is considered as cachexia clinically [33], we started the administration of WAF, EtAF, and AICAR from day 9. As shown in Figure 2A, the administration of WAF and EtAF significantly suppressed weight loss induced by cancer, compared to the tumor group. However, both extracts’ administration did not change tumor volume and weight (Figure 2B,C,F). These results indicated that the weight loss inhibitory effect of WAF and EtAF was not due to the reduction in tumor size. The weight changes have a more significant difference after the tumor weights were subtracted from the total body weight. The tumor group lost 18.90% weight compared to the control group, whereas WAF, EtAF, and AICAR administrated groups only lost 12%, 12.61%, and 14.23% of their weights, respectively, compared to the control group (Figure 2D). During the experiment, the administration of the extracts did not affect food intake (Figure 2E).

### 3.3. AF Treatment in Mild Cachectic Mice Regulated Atrophy of Muscle and WAT by Reducing Serum IL-6

To elucidate the anti-cachectic mechanism of WAF and EtAF, we confirmed serum IL-6 levels. As shown in Figure 3A, serum IL-6 was increased in tumor-bearing mice. However, WAF and EtAF significantly reduced IL-6 secretion. Compared to the tumor group, WAF and EtAF also prevented the wasting of muscle and adipose tissue, which is a similar effect to AICAR (Figure 3B,C). However, there was no significant weight increase in the heart, which is the muscular organ [34] (Figure 3D). Next, the reduction of gMuscle atrophy was confirmed by histological experiments. The muscle fiber of both extract-administrated groups were thicker than that of the tumor group (upper layer of Figure 3E). It was also confirmed that the reduction in adipocytes size of eWAT was prevented by the administration of WAF, EtAF, and AICAR (lower layer of Figure 3E). The adipocyte size of the tumor group was 402 ± 119 μm^2^, but those of the groups administered with WAF and EtAF were 2071 ± 387 μm^2^ and 1870 ± 654 μm^2^, respectively. The adipocyte size of the AICAR group was 2399 ± 1055 μm^2^ (Figure 3F). In addition, the number of adipocytes per field was significantly reduced in both extracts and AICAR-treated mice, confirming an enlarged adipocyte size compared to the tumor group (Figure 3G).

### 3.4. AF Treatment in Mild Cachectic Mice Regulated Muscle Degradation Factors and WAT Differentiation Factors

mRNA expression levels of the muscle degradation factor, MAFbx and MuRF-1, were decreased in WAF and EtAF groups compared to the tumor group (Figure 4A). As with the mRNA results, increased protein levels of MAFbx and MuRF-1 were decreased by the administration of both extracts (Figure 4B,C). In addition, we confirmed whether these extracts could improve adipose tissue wasting in the cachexia model. The adipocyte differentiation factors PPARγ and C/EBPα were reduced by the tumor compared to the control group. However, the administration of both extracts suppressed this reduction (Figure 4D–F). Browning of white adipocyte is a typical phenomenon observed in cancer-induced cachexia [35]. It has been reported that the inactivation of AMPK induces browning and eventually leads to loss of adipose tissue [36,37]. In this experiment, browning marker UCP1 expression was increased, and AMPK was inactivated in the tumor group. Administration of WAF and EtAF can suppress the elevation of UCP1 and activate AMPK (Figure 4D–F).

### 3.5. WAF Treatment in Severe Cachectic Mice Improved Tumor-Induced Weight Loss

To mimic the diffuse carcinomatosis, mice were intraperitoneally injected with CT-26 cells. Since the tumor group showed a decreased body weight of approximately 10% on day 7 after CT-26 cell inoculation, we started the administration of WAF and EtAF from day 7. During the 16 days, the weight of the control group increased by 21.23% from the starting weight, while the tumor group and EtAF group increased only by 3.93% and 3.90%, respectively. However, the WAF group significantly increased their weight by 12.46% (Figure 5A). Mortality results showed a similar tendency to weight loss. The survival rates of tumor, WAF, and EtAF groups were 23.94%, 52.76%, and 34.2%, respectively. The mortality of the cancer-cachexia mouse model was decreased by both extracts’ administration. Although the weight loss was not suppressed in the EtAF group, the EtAF group showed a 10% higher survival rate than the tumor group (Figure 5B). During this period, the food intake of WAF and EtAF did not show a difference compared to the tumor group (Figure 5C). After sacrifice, we measured the weight of muscle, including gMuscle and heart. When compared to the tumor group, administration of both extracts did not improve not only atrophy of gMuscle and heart but also tumor weight (Figure 5D–F).

### 3.6. AF Treatment in Severe Cachectic Mice Prevents Atrophy of Adipose Tissue

In the severe cachectic mice model, eWAT weight of the tumor group was significantly reduced compared to the control group. However, the weight loss of eWAT in the WAF and EtAF groups was less than that of the tumor group (Figure 6A). In addition, the adipocyte size of the tumor group was reduced compared to the control group (Figure 6B,C). In the tumor group, much smaller adipocytes than other adipocytes were observed, which appears to be a browning phenomenon. However, the administration of both extracts prevented atrophy of adipose tissue and browning phenomenon. Similar to the mildly induced cachexia model, a decreased number of adipocytes per field in WAF- and EtAF-treated mice indicated increased adipocyte size compared to the tumor group (Figure 6D). These results suggest that WAF and EtAF can inhibit atrophy of WAT.

### 3.7. AF Regulated C2C12 Myoblast Proliferation and Degradation Factors in CT-26 CM-Treated Condition

Decreased cell viability interferes with the regeneration of muscle cells, leading to muscle atrophy [38]. To confirm whether WAF and EtAF affect the cell viability of C2C12 cells, WAF and EtAF were directly administered to C2C12 cells. As shown in Figure 7A, the cell viability was not affected by WAF and EtAF at concentrations of up to 100 μg/mL. However, the C2C12 cells treated with CM showed a decreased cell proliferation and the decrease of cell viability by CM was inhibited by WAF-CM and EtAF-CM (Figure 7B). Since IL-6 affected muscle atrophy in our in vivo model, we also confirmed the IL-6 production from CT-26 cells. As expected, IL-6 secretion was decreased in the WAF-CM and EtAF-CM (Figure 7C). These results indicated that WAF and EtAF did not directly affect the cell proliferation of C2C12 cells, but they could reduce the cancer cell-mediated growth inhibition by decreasing IL-6 secretion from cancer cells. It has been reported that the treatment of CM decreased the proliferation of C2C12 myoblast by controlling cell cycle-related factors [23]. Similar to reported studies, treatment of CM significantly increased the p21 mRNA level and reduced mRNA levels of CDK2 and cyclin D in C2C12 cells. However, WAF-CM and EtAF-CM inhibited the changes of cell cycle-related factors, which were altered by CM, except cyclin D treated with EtAF-CM (Figure 7D). We also confirmed that CM from CT-26 cells increased the degradation factors, MAFbx and MuRF-1, in the process of C2C12 differentiation into myotube. However, this upregulation was suppressed in WAF-CM and EtAF-CM groups. Meanwhile, a myotube maker MyH [39] was increased by WAF-CM and EtAF-CM compared to CM treated group (Figure 8).

### 3.8. AF Increased 3T3-L1 Differentiation in CT-26 CM-Treated Condition

Adipocyte differentiation was inhibited by recombinant cytokines, including TNF-α, IL-6, and IL-1β, and IL-6 exerted the most inhibitory effect [40]. First, we confirmed whether CM reduces the differentiation of adipocytes. The treatment of CM significantly reduced the differentiation of adipocytes compared to the control group. However, WAF-CM and EtAF-CM suppressed this reduction (Figure 9A,B). Similar to the C2C12 viability result, the treatment of CM affected differentiation ability without a decrease of cell viability (Figure 9C). In addition, the expression levels of adipocyte differentiation factors, PPARγ, C/EBPα, fatty acid synthase (FAS), lipoprotein lipase (LPL), and adipocyte binding protein 2 (aP2), were reduced in the CM treatment group. However, this reduction was suppressed in the WAF-CM- and EtAF-CM-treated groups (Figure 9D–F).

## 4. Discussion

Currently, the pathogenesis of cancer cachexia has not been clearly elucidated, and there are only a few FDA-approved drugs such as progestagens [41]. Although clear mechanisms have not been elucidated yet, systemic inflammation is still an important therapeutic target [4]. Corticosteroids and anabolic androgenic steroids as well as drugs targeting systemic inflammation are used in cancer patients. However, their use is limited due to side effects [42]. Several studies have shown that it is useful to combine more than one drug [43]. Therefore, using extracts of natural products containing various ingredients can be one option for improving cachexia induced by cancer. In our previous study, AF reduced allergic inflammation by suppressing histamine and pro-inflammatory cytokines, including IL-6, IL-8, IL-1β, and TNF-α in vitro and in vivo [21]. In addition, AF not only inhibited the growth of colon cancer cells but also regulated body weight through phosphorylation of AMPK, which is an energy homeostasis-related factor [19,20]. Therefore, we expected that AF could show an anti-cachectic effect.

Although IL-6 is not considered the sole inducer of cachexia, an increased level of IL-6 is a typical symptom found in cancer patients and is one of the major factors in the development of cancer cachexia [44]. MAFbx and MuRF-1 are muscle-specific E3 ubiquitin ligases expressed in skeletal muscle. Expressions of MAFbx and MuRF-1 are increased in atrophy-inducing conditions such as cachexia [8]. Yakabe et al. reported that systemic IL-6 was related to increased expression of MuRF1 in a disuse-induced muscle atrophy mice model [45]. In an Apc(Min/+) mouse, circulating IL-6 was necessary for MAFbx. In a wasting condition, MAFbx was expressed by IL-6 [46]. We injected CT-26 cells subcutaneously into BALB/c mice to induce cachexia. This model has been commonly used in cachexia studies due to their short induction period. As with other muscle atrophy mice models [45,46], IL-6 was increased by the injection of CT-26 cells and muscle atrophy was also reproduced [23,47]. In our study, injection of CT-26 cells induced an increase in IL-6, which induced an increase of MAFbx and MuRF-1 in muscle, resulting in muscle atrophy. However, the administration of WAF and EtAF reduced the serum IL-6 level (Figure 3A). Similar to previous reports [45,46], muscle degradation factors, MAFbx and MuRF-1, were suppressed in IL-6 decreased mice groups. These groups were administrated with WAF and EtAF, indicating that WAF and EtAF reduced muscle atrophy by reducing circulating IL-6 (Figure 3B,E, Figure 4A–C). Moreover, IL-6 secreted from CT-26 cells was reduced by treating WAF and EtAF in vitro (Figure 7C). The muscle degradation factors, MAFbx and MuRF-1, were increased in C2C12 cells cultured with CM. However, the increase of muscle degradation factors was suppressed in C2C12 cells cultured with WAF-CM and EtAF-CM (Figure 7 and Figure 8). Therefore, these results suggested that the treatment of WAF and EtAF inhibited muscle degradation through IL-6 reduction.

Loss of adipose tissue is associated with shorter survival time for advanced cancer patients [48]. Specific genes related to adipogenesis and lipogenesis are changed in adipose tissue of cachexia condition, inducing atrophy of adipose tissue [15]. During adipogenesis and lipogenesis, PPARγ, C/EBPα, and aP2, which are adipogenic-specific transcription factors, play a crucial role in adipocyte formation and lipid production [49,50,51]. Furthremore, FAS and LPL, which are lipogenic enzymes associated with lipogenesis, induce fatty acid synthesis and lipid accumulation [52]. These factors are confirmed in obesity research as molecular markers to measure the adipocyte differentiation [53,54,55]. Baltgalvis et al. reported that IL-6 could accelerate the loss of fat [46]. In our experiments, IL-6 was increased in the tumor group and showed a decreased fat amount, compared to healthy mice. However, the fat amount and expression level of adipocyte differentiation factors, PPARγ and C/EBPα, were restored by the administration of WAF and EtAF (Figure 3, Figure 4D–F). Our in vitro results showed a similar result with in vivo results. The adipogenic factors were decreased in 3T3-L1 cells differentiated with CM. However, the decrease of adipogenic factors was restored in 3T3-L1 cells cultured with WAF-CM and EtAF-CM (Figure 9). Therefore, these results suggest that the treatment of WAF and EtAF increased adipogenesis and lipogenesis.

As a result, the administration of WAF and EtAF suppressed weight loss by inhibiting the atrophy of muscle and adipose tissue. Although patients suffering from cachexia eat sufficient amounts of food, they still experience uncontrolled weight loss due to systemic inflammation [3]. During this period, food intake was not affected by WAF and EtAF (Figure 2). In addition, the administration of WAF and EtAF did not show drug toxicity. Instead, it improved the hepatotoxicity and nephrotoxicity induced by cachexia (Table 2). These results indicated that WAF and EtAF suppressed weight loss by regulating systemic inflammation without additional nutrition. Since cachexia is originally induced by cancer, tumor size affects the severity of cachexia induced by the tumor [56]. There was no change in tumor volume and weight by the administration of WAF and EtAF compared to the tumor group (Figure 2B,C). These results showed that our drugs could be cachexia-targeting drugs in cancer cachexia conditions. It is also expected that the anti-cachectic effect of WAF and EtAF will be better when administered with anti-cancer drugs, so further studies of the concomitant administration with anti-cancer drugs are also needed. Our severely cachectic mice model confirmed that loss of weight and appetite eventually induced cancer death. These results were consistent with other studies, which reported that mortality was closely related to weight loss [57]. However, when comparing to the tumor group, the administration of WAF and EtAF reduced mortality by 20% and 10%, respectively (Figure 5). Our results proved that the increased weight of muscle and adipose tissue by the administration of WAF and EtAF lowered mortality by inhibiting cachexia symptoms.

Another insight of this paper is that AF reduced the expression level of UCP1 by inducing AMPK activity in adipose tissue. Increased UCP1 in white adipose tissue is a representative browning phenomenon. Through browning, cancer cells consume more energy and exacerbate cachexia symptoms [57]. In cachexia, browning, which is normally induced by activated AMPK, was paradoxically induced by an inactivated AMPK [36]. In our study, the adipocyte size of the cachexia-induced group was much smaller than that of the control group (Figure 3F and Figure 6C). Brown adipocytes are characterized by a very small adipocyte size compared to white adipocytes [58]. Therefore, we determined that the tumor group showed browning and the administration of WAF and EtAF suppressed this phenotypic change. In addition, the administration of WAF and EtAF activated AMPK, which was inactivated due to cachexia. As with the previous report [36], the expression level of UCP1 was decreased in the AMPK-activated group (Figure 4E). Through this mechanism, unnecessary energy consumption was suppressed, and as a result, the atrophy of adipose tissue was decreased.

## 5. Conclusions

The first significant point of our study is that we demonstrated the anti-cachectic effect of AF. AF prevented wasting of muscle and adipose tissue by suppressing systemic inflammation in the mice model. In particular, AF improved the browning phenomenon in adipose tissue by restoring the activation of AMPK and increased the survival rate by inhibiting weight loss. In addition, AF is a non-toxic functional food that can be used as a tea. For a patient of 70 kg, the dose of 100 mg/kg/day used in the mice experiment means 7 g of WAF or EtAF per day. This simple calculation shows that 100 g and 50 g of AF are required for clinical administration of WAF and EtAF, respectively. Even though a dose of 100 mg/kg/day did not induce hepatotoxicity and nephrotoxicity in the mice model, further studies for appropriate doses are needed for clinical use. Therefore, we expect that cancer patients could use it as a therapeutic supplement without worrying about side effects.

## Figures and Tables

**Figure 1 nutrients-12-03195-f001:**
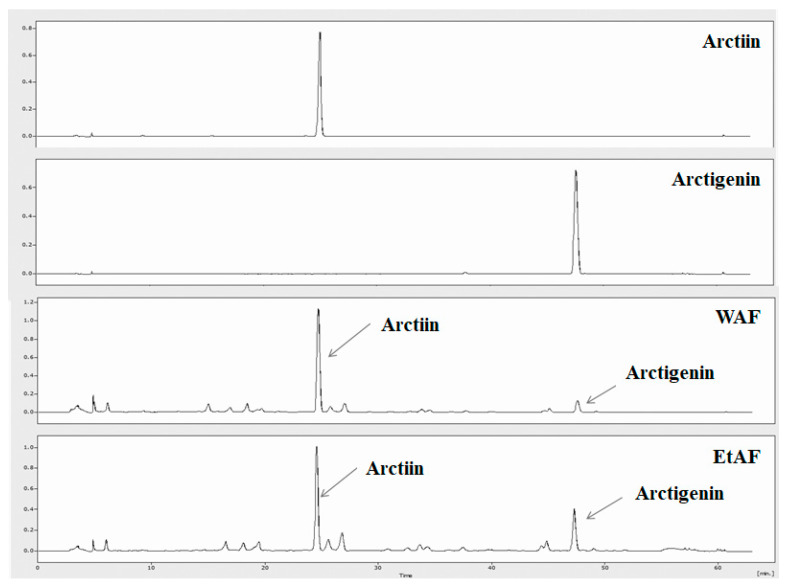
HPLC of major compounds of Arctii Fructus (AF). The chromatographic peaks of arctiin, arctigenin, water extract of AF (WAF), and ethanol extract of AF (EtAF).

**Figure 2 nutrients-12-03195-f002:**
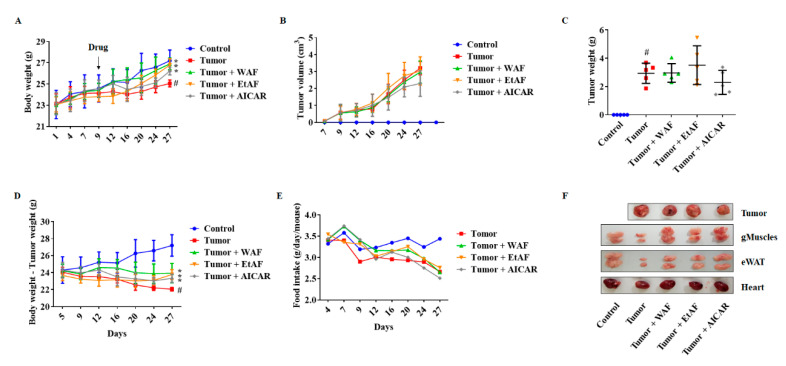
Effects of AF on the cachectic symptom in CT-26 tumor-bearing mice. CT-26 cells were subcutaneously injected into the abdominal area of mice (*n*  =  5). 100 mg/kg/day of WAF and EtAF were orally administrated every day, and 250 mg/kg of 5-aminoimidazole-4-carboxamide-1-β-D-ribofuranoside (AICAR) was intraperitoneally injected every 2 days. AICAR was used as a positive control. Weight changes for 27 days (**A**) and tumor volume changes (**B**) were measured on the indicated day. The tumor weights were measured after sacrificing the mice (**C**). Body weight-tumor weight (**D**) and food intake (**E**) were measured on the indicated day. The representative images of the tumor, gMuscle, eWAT, and heart were captured (**F**). AICAR was used as a positive control. All values are mean ± SD. # *p* < 0.05, significantly different from control group; * *p* < 0.05, significantly different from tumor group.

**Figure 3 nutrients-12-03195-f003:**
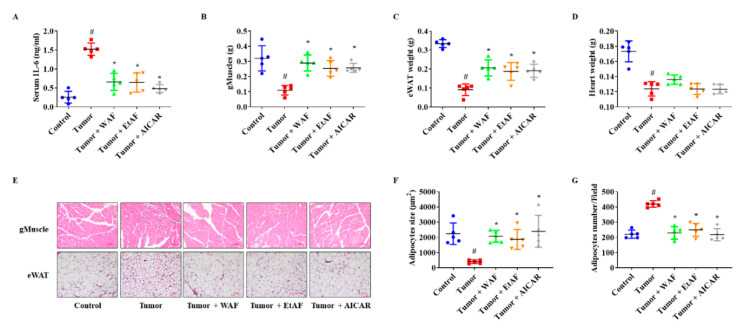
Effects of AF in serum IL-6, muscle weight, and adipose tissue from CT-26 tumor-bearing mice. The serum IL-6 level was determined after sacrificing the mice (**A**). The weights of gMuscle (**B**), eWAT (**C**), and heart (**D**) were measured after sacrificing the mice. The H&E staining images of gMuscle and eWAT. The magnitude is ×200 and the scale bar is 100 µm (**E**). H&E-stained areas of eWAT were measured using ImageJ (**F**). Adipocytes numbers were counted by randomly selecting fields (**G**). All values are mean ± SD. # *p* < 0.05, significantly different from control group; * *p* < 0.05, significantly different from tumor group.

**Figure 4 nutrients-12-03195-f004:**
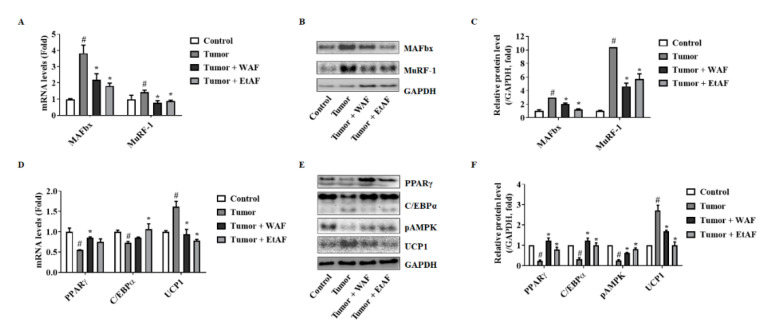
Effects of AF in gMuscle and eWAT from CT-26 tumor-bearing mice. mRNA levels of MAFbx and MuRF-1 in gMuscle (**A**). The proteins levels of MAFbx and MuRF-1 (**B**) and relative levels (**C**) in gMuscle. The mRNA expression levels of PPARγ, C/EBPα, and UCP1 in eWAT (**D**). The protein levels of PPARγ, C/EBPα, pAMPK, and UCP1 in eWAT (**E**). The relative protein expression levels of eWAT (**F**). GAPDH was used as an endogenous control. All values are mean ± SD. # *p* < 0.05, significantly different from control group; * *p* < 0.05, significantly different from tumor group.

**Figure 5 nutrients-12-03195-f005:**
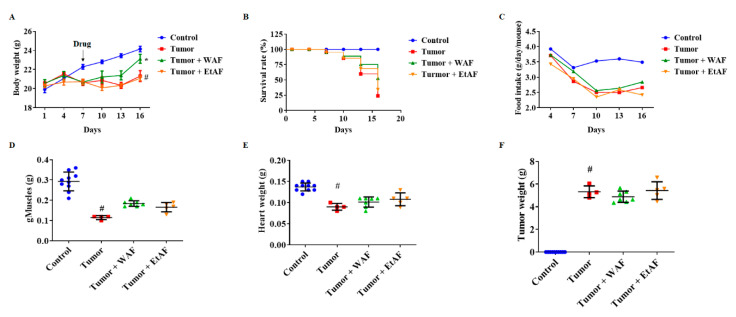
Effects of AF on mortality and cachectic symptom in intraperitoneally CT-26 tumor-bearing mice. CT-26 cells were intraperitoneally injected into the mice (*n* = 10). 100 mg/kg/day of WAF and EtAF were orally administrated every day. Weight changes for 16 days were measured on the indicated day (**A**). Survival rate (**B**) and food intake (**C**) were measured on the indicated day. The weights of gMuscle (**D**), heart (**E**), and tumor (**F**) were measured after sacrificing the mice. All values are mean ± SD. # *p* < 0.05, significantly different from control group; * *p* < 0.05, significantly different from tumor group.

**Figure 6 nutrients-12-03195-f006:**
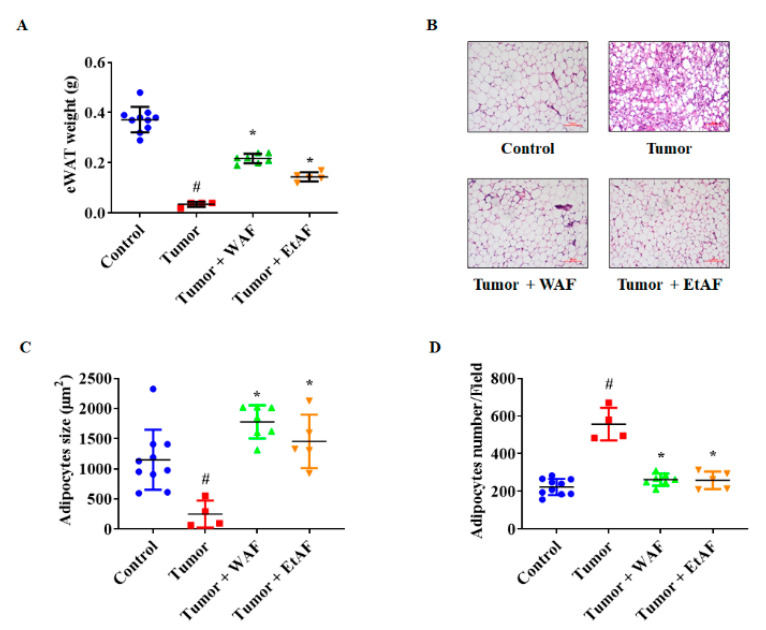
Effects of AF on eWAT wasting in intraperitoneally CT-26 tumor-bearing mice. The weight of eWAT was measured after sacrificing mice (**A**). The H&E staining images of eWAT (**B**). The magnitude is ×200 and the scale bar is 100 µm. Adipocyte size was measured using ImageJ (**C**). The adipocyte number of each group was counted by randomly selecting fields (**D**). All values are mean ± SD. # *p* < 0.05, significantly different from control group; * *p* < 0.05, significantly different from tumor group.

**Figure 7 nutrients-12-03195-f007:**
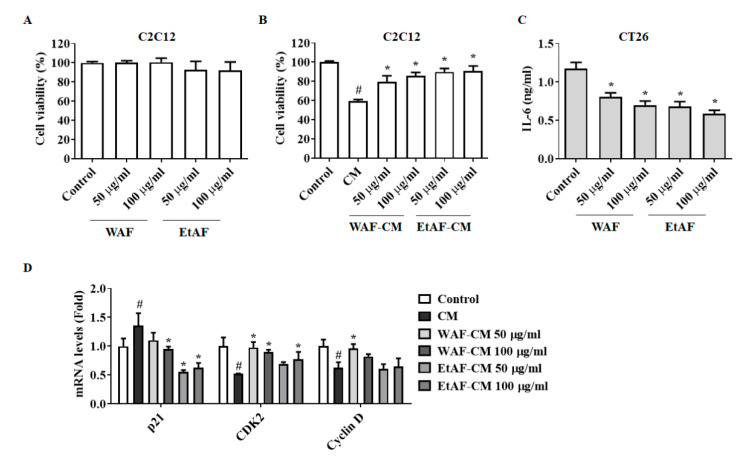
Effects of AF on cell viability and cell cycle arrest of C2C12 myoblast treated with conditioned media (CM) from CT-26 cells. C2C12 myoblasts were treated with WAF and EtAF for 48 h, and then cell viability was determined (**A**). C2C12 myoblasts were incubated with CM treated with or without WAF and EtAF after 50% dilutions in differentiation media. After 36 h, cell viability was measured (**B**). IL-6 secretion from WAF and EtAF-treated CT-26 cells (**C**). Using CM-treated C2C12 cells, qPCR and Western blot were conducted. mRNA levels of cell cycle related factors were measured (**D**). GAPDH was endogenous control. Results are from three independent experiments. All values are mean ± SD. # *p* < 0.05, significantly different from control group; * *p* < 0.05, significantly different from CM group.

**Figure 8 nutrients-12-03195-f008:**
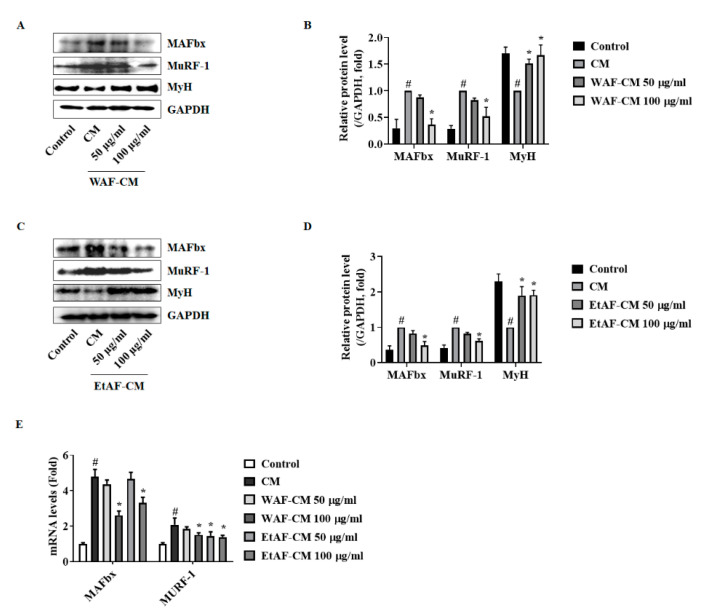
Effects of AF on C2C12 myoblast treated with conditioned media (CM) from CT-26 cells. Protein levels of MAFbx, MuRF-1, and MyH were measured after treating WAF-CM (**A**). The relative protein expression levels of MAFbx, MuRF-1, and MyH (**B**). Protein levels of MAFbx, MuRF-1, and MyH were measured after treating with EtAF-CM (**C**). The relative protein expression levels (**D**). GAPDH was endogenous control. mRNA levels of MAFbx and MuRF-1 were measured (**E**). GAPDH was used as an endogenous control. The results are from three independent experiments. All values are mean ± SD. # *p* < 0.05, significantly different from control group; * *p* < 0.05, significantly different from CM group.

**Figure 9 nutrients-12-03195-f009:**
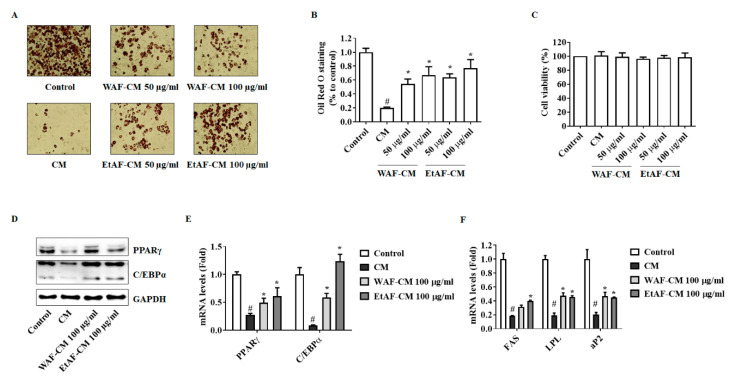
Effects of AF on the differentiation of 3T3-L1 cell treated with conditioned media (CM) from CT-26 cells. Oil Red O staining images (200× magnification) (**A**). Stained Oil Red O solution was resolved and measured (**B**). Cell viability was determined after incubating with CM treated with or without WAF-CM and EtAF-CM (**C**). The protein and mRNA levels of PPARγ and C/EBPα in 3T3-L1 cells after fully differentiation (**D**,**E**). GAPDH was used as an endogenous control. mRNA levels of FAS, LPL, and aP2 were measured (**F**). The results are from three independent experiments. All values are mean ± SD. # *p* < 0.05, significantly different from control group; * *p* < 0.05, significantly different from CM group.

**Table 1 nutrients-12-03195-t001:** Primer sequences for real-time RT-PCR.

Target Gene	Primer Sequences
mPPAR	5′-TTTCAAGGGTGCCAGTTTC-3′ (sense)
	5′-TTATTCATCAGGGAGGCCAG-3′ (antisense)
mC/EBPα	5′-GCCGAGATAAAGCCAAACAA-3′ (sense)
	5′-CGTAAATGGGGATTTGGTCA-3′ (antisense)
mLPL	5′-TGCCGCTGTTTTGTTTTACC-3′ (sense)
	5′-TCACAGTTTGCTGCTCCCAGC-3′ (antisense)
mFAS	5′-TGGTGGGTTTGGTGAATTGTC-3′ (sense)
	5′-GCTTGTCCTGCTCTAACTGGAAGT-3′ (antisense)
maP2	5′-CGTAAATGGGGATTTGGTCA-3′ (sense)
	5′-TCGACTTTCCATCCCACTTC-3′ (antisense)
mUCP1	5′-TCGACCTTAAAGGAATCCCC-3′ (sense)
	5′-CACAGGCTTTCCTTCTTTGC-3′ (antisense)
mMAFbx	5′-TCACAGCTCACATCCCTGAG-3′ (sense)
	5′-GACTTGCCGACTCTCTGGAC-3′ (antisense)
mMuRF-1	5′-ATGGAGAACCTGGAGAAGCA-3′ (sense)
	5′-ACGGTCCATGATCACCTCAT-3′ (antisense)
mGAPDH	5′-AACTTTGGCATTGTGGAAGG-3′ (sense)
	5′-GGATGCAGGGATGATGTTCT-3′ (antisense)

**Table 2 nutrients-12-03195-t002:** Cytotoxicity of WAF and EtAF on serum levels in mildly induced cachexia mice.

	AST (IU/L)	ALT (IU/L)	Creatinine (mg/dL)	BUN (mg/dL)
Control	94.60 ± 12.11	27.60 ± 6.65	0.14 ± 0.02	18.10 ± 2.42
Tumor	388.70 ± 81.43 ^#^	139.40 ± 16.15 ^#^	0.23 ± 0.03 ^#^	31.50 ± 4.09 ^#^
Tumor + WAF	157.90 ± 23.08 *	72.00 ± 8.47 *	0.17 ± 0.04 *	23.20 ± 2.20 *
Tumor + EtAF	166.20 ± 31.70 *	71.50 ± 14.36 *	0.18 ± 0.04 *	23.10 ± 3.00 *

All values are mean ± SD. ^#^
*p* < 0.05, significantly different from control group; * *p* < 0.05, significantly different from tumor group.

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
