# Peer review of "The Extract of Arctium lappa L. Fruit (Arctii Fructus) Improves Cancer-Induced Cachexia by Inhibiting Weight Loss of Skeletal Muscle and Adipose Tissue"

_nutrients, 2020, doi:10.3390/nu12103195_

Round 1
Reviewer 1 Report
Paper: Arctii Fructus ameliorates cancer-induced cachexia by inhibiting weight loss of skeletal muscle and adipose tissue
The present work deals with a very interesting problem that occurs during the development of most of the cancers described, such as cachexia, understood in terms of loss of muscle and fat mass. The authors study in the present work the role of Arctii fructus, the dried fruit of Arctium lappa L, widely used in traditional oriental medicine.
As I said, the work seems interesting to me due to the subject discussed and the objective (s) proposed by the authors are well established, however, for this work to be published in Nutrients, the authors must carry out a major and important review of it.
1.- Title:
I believe that the title should be modified since I suspect that the authors use an extract of the fruit and not the fruit directly, in addition and for a better understanding for readers, the authors should introduce the species from which these fruits come (Arctium lappa L). Also I am not sure that the term "ameliorates" is very appropriate for the title of a scientific work.
2.-Plant extracts:
Papers sent to journals in the scientific category of Nutrients, which use extracts and study their biological effects, require the authors to carry out a detailed study of the composition of said extracts as well as their proportions in order to be able to identify the activity with the exact composition of the same. Therefore, it is not valid that this aspect is solved by putting two lines in the text on the composition taken from other sources.
For this reason, the authors of this work must include in it what is requested in the previous paragraph and thus be able to continue with the peer review since from a methodological point of view, the authors have carried out a very appropriate work and therefore it is absolutely necessary that we know the true composition of the extracts used. In this case, it is not valid that the authors include at the end a simple HPLC study on the composition where only the existence of two components arctiin and arctigenin are observed and it is not said to which group of compounds they belong. That is not possible, there must be many more with possible anti-cachetic activity.
3.-Materials and Methods:
2.3. Cachexia mice model induced by colorectal cancer cell. As this section is described, it is very difficult for any reader to follow the model described by the authors. It appears in the text first 9 days, later 7 days. A medium cachexia is mentioned first, and later it is not known if a strong cachexia is also induced ... etc. Authors should rewrite that section and include a table of times and treatments that improves understanding of this section. By the way, in this section the acronyms AICAR appear that we do not know why they are because they are not clarified in the manuscript.
I understand that CT26 cells is a murine colorectal carcinoma cell line which is from a BALB / c mouse, I ask why is it possible to obtain a cachexia situation when those mice are injected with cells of themselves? This needs to be explained.
2.5. CT-26 conditioned media (CM). This section is also confusing since as far as I understand it is exclusively about the culture medium for cells to grow and divide. If not, the authors should clarify what we have to understand by “conditioned media” and what its true objective is.
4.-Results:
This section is essential and everything should be very clear. Throughout the manuscript, the authors “abuse” the use of acronyms greatly and that makes understanding the work very difficult. It is true that currently we must use them but in this case it is overused. Therefore, the authors must make a great effort to reduce them as much as possible. In many cases they must define the situation instead of using the corresponding acronyms. And of course acronyms should not appear without being properly defined.
The figures presented by the authors are heavily loaded with data for a journal format that makes the results presented unclear in the end. For this reason, the authors should redistribute the results in two or three more figures and pass figure 7 as well as the complete composition analysis they carry out in the complementary results section.
The acronyms that precede the elements that are studied should go at the beginning and not at the end of each situation. In every possible case, the authors should overwrite the situation they are studying on each figure, for example, muscle weight, heart weight, etc.
It is not clear why the authors use * and # in the statistical differences. It should be clarified in each case and in section 2.14. Statistical analysis.
5.- Discussion:
The authors make a very succinct discussion for the large number of results they present. For this reason, they must rewrite it, placing special emphasis on each of the results shown in the manuscript. There are many markers that have been studied to make a very simple and rudimentary discussion.
Author Response
- Title:
I believe that the title should be modified since I suspect that the authors use an extract of the fruit and not the fruit directly, in addition and for a better understanding for readers, the authors should introduce the species from which these fruits come (Arctium lappa L). Also I am not sure that the term "ameliorates" is very appropriate for the title of a scientific work.
→ As you commented, we edited the title (Line 2).
- Plant extracts:
Papers sent to journals in the scientific category of Nutrients, which use extracts and study their biological effects, require the authors to carry out a detailed study of the composition of said extracts as well as their proportions in order to be able to identify the activity with the exact composition of the same. Therefore, it is not valid that this aspect is solved by putting two lines in the text on the composition taken from other sources.
For this reason, the authors of this work must include in it what is requested in the previous paragraph and thus be able to continue with the peer review since from a methodological point of view, the authors have carried out a very appropriate work and therefore it is absolutely necessary that we know the true composition of the extracts used. In this case, it is not valid that the authors include at the end a simple HPLC study on the composition where only the existence of two components arctiin and arctigenin are observed and it is not said to which group of compounds they belong. That is not possible, there must be many more with possible anti-cachetic activity.
→ Arctiin and arctigenin are the main bioactive compounds of AF [29]. It has been reported that arctigenin and arctiin account for approximately 0.5-2% (w/w) and 2-10% (w/w) of the dry weight of the fruit, respectively [Gao, Q.; Yang, M.; Zuo, Z. Overview of the anti-inflammatory effects, pharmacokinetic properties and clinical efficacies of arctigenin and arctiin from Arctium lappa L. Acta. Pharmacol. Sin. 2018, 39, 787-801, doi: 10.1038/aps.2018.32.]. These compounds also possess anti-inflammatory effect. Arctiin can inhibit pro-inflammatory mediators such as IL-1β, IL-6, TNF-α, and PGE2 through inactivation of NF-κB. [Lee, S.; Shin, S.; Kim, H.; Han, S.; Kim, K.; Kwon, J.; Kwak, J.H.; Lee, C.K.; Ha, N.J.; Yim, D.; Kim, K. Anti-inflammatory function of arctiin by inhibiting COX-2 expression via NF-κB pathways. J. Inflamm. (Lond). 2011, 8, 16, doi: 10.1186/1476-9255-8-16.]. In addition, arctigenin shows inhibitory effect on inflammation by reducing IL-6, TNF-α, NO, and COX-2 expression in activated macrophages [Feng Zhao, Lu Wang, Ke Liu, In vitro anti-inflammatory effects of arctigenin, a lignan from Arctium lappa L., through inhibition on iNOS pathway. J Ethnopharmacol. 2009 Apr 21;122(3):457-62. doi: 10.1016/j.jep.2009.01.038.].
- Materials and Methods:
2.3. Cachexia mice model induced by colorectal cancer cell. As this section is described, it is very difficult for any reader to follow the model described by the authors. It appears in the text first 9 days, later 7 days. A medium cachexia is mentioned first, and later it is not known if a strong cachexia is also induced ... etc. Authors should rewrite that section and include a table of times and treatments that improves understanding of this section. By the way, in this section the acronyms AICAR appear that we do not know why they are because they are not clarified in the manuscript.
→ We explained animal models in more detail (Materials and Methods 2.3. section). To improve the understanding of animal models, we added in supplementary figure. Also, we added full name of AICAR (2.1. Reagents and antibodies). In this study, we used AICAR as a positive drug for cachexia. This is mentioned in Material and Methods section (2.3).
I understand that CT26 cells is a murine colorectal carcinoma cell line which is from a BALB / c mouse, I ask why is it possible to obtain a cachexia situation when those mice are injected with cells of themselves? This needs to be explained.
→ As you commented, CT26 cells is a murine colorectal carcinoma cell line which is from a BALB / c mouse. We injected CT-26 cells subcutaneously into BALB/c mice to induce cachexia. Even though we didn’t experiment cachexia inducing mechanism, this model has been commonly used in cachexia studies due to their short induction period (Kim et al., Sci Rep 2016; Choi et al., Mediators Inflamm 2014).
There are also several mice model. Yakabe et al. reported that systemic IL-6 was related to increased expression of MuRF1 in disuse-induced muscle atrophy mice model (Yakabe et al., plos one, 2018). In an Apc(Min/+) mouse, circulating IL-6 was necessary for MAFbx. In a wasting condition, MAFbx was expressed by IL-6 (Baltgalvis et al., Pflugers Arch, 2009).
As with other muscle atrophy mice models (Yakabe et al., plos one, 2018; Baltgalvis et al., Pflugers Arch, 2009), IL-6 was increased by the injection of CT-26 cells and muscle atrophy was also reproduced (Kim et al., Sci Rep 2016 ;Choi et al., Mediators Inflamm 2014). For this reason, we think IL-6 is related to cachexia inducing in this model. We added this point in Discussion section (Line 395-406).
2.5. CT-26 conditioned media (CM). This section is also confusing since as far as I understand it is exclusively about the culture medium for cells to grow and divide. If not, the authors should clarify what we have to understand by “conditioned media” and what its true objective is.
→ Conditioned media (CM) is spent media harvested from cultured cells. So, CM is normal growth media (10% FBS/DMEM) for CT-26 cells. To mimic the cancer status administered with WAF and EtAF, CT-26 cells cultured media treated with WAF and EtAF were used in in vitro as CM for 3T3-L1 cells and C2C12 cells. Kim et al (Sci Rep, 2016) made CT-26 CM after treating Citrus unshiu peel extract. And then, they also treated these CM into C2C12 cells. We prepared CT-26 CM according to the previous report with slight modifications. As you commented, we described it more clearly in Materials & Methods 2.5. section.
- Results:
This section is essential and everything should be very clear. Throughout the manuscript, the authors “abuse” the use of acronyms greatly and that makes understanding the work very difficult. It is true that currently we must use them but in this case it is overused. Therefore, the authors must make a great effort to reduce them as much as possible. In many cases they must define the situation instead of using the corresponding acronyms. And of course acronyms should not appear without being properly defined.
→ As your comment, Result section was edited to clear the our experiment results.
The figures presented by the authors are heavily loaded with data for a journal format that makes the results presented unclear in the end. For this reason, the authors should redistribute the results in two or three more figures and pass figure 7 as well as the complete composition analysis they carry out in the complementary results section.
→ As your comment, the Figures were divided and relocated.
The acronyms that precede the elements that are studied should go at the beginning and not at the end of each situation. In every possible case, the authors should overwrite the situation they are studying on each figure, for example, muscle weight, heart weight, etc.
It is not clear why the authors use * and # in the statistical differences. It should be clarified in each case and in section 2.14. Statistical analysis.
→ We described meaning of * and # in every figure legends. Also, as you commented, we added use * and # in section 2.14. Statistical analysis.
- Discussion:
The authors make a very succinct discussion for the large number of results they present. For this reason, they must rewrite it, placing special emphasis on each of the results shown in the manuscript. There are many markers that have been studied to make a very simple and rudimentary discussion.
- As you commented, we edited overall discussion to describe the results.

Reviewer 2 Report
The paper by Yo-Han Han and coworkers aims to describe the beneficial effect of Arctii Fructus on cancer –induced cachexia. The authors used in vivo and in vitro experiments. The main result of this study is that Arctii Fructus demonstrated anti-cachectic effects.
In general, the manuscript is well written, however there are several serious questioning concerning bibliography and Methods.
MAJOR COMMENTS
1) In the introduction
- the authors had forgotten that skeletal muscle loss is part of the parameters that reduce efficacy of anti-cancer treatments (Sjoblom B Lung Cancer 2015, Degens 2019) line 43.
- The authors used the reference of Gullett et al. to indicate that “approximately 20% of them died directly because of cachexia” (lines 46-47). It is interesting since Gullett et al cited Inui et al 2002 and MacDonald et al. 2003. Inui and his collaborators cited Bruera E. 1997. With all the respect that I have for Eduardo Bruera, this review did not provide any data on mortality related to cancer cachexia. Concerning Mc Donalds’ article I did not find data on mortality too. The authors need to find the adequate reference if they want to argue about direct effect of cachexia on patients’ mortality. Be careful, please read the articles!
- the authors claim that in cancer patient MuRF1 and MAFbx are significantly increased and then can be considered as markers of muscle atrophy ref [7] line 51-52. Again, the authors should have read the review of Bodine SC, since data reported on Mafbx expression in muscle during cancer cachexia extracted from preclinical model. Again, the authors must use the appropriate reference by looking carefully to the literature, since there is no really consensus on MuRF1 and MAFbx increase in cancer patient skeletal muscle.
- Line 58, the authors cite Shachar S. Again the authors need to read very carefully this article. In fact (see Table 2) on 4 clinical studies 2 concluded that high SAT is associated with worse survival, and one (Antoun S et al.) concluded that “It seems that a high volume of SAT could be a metabolic hallmark of less aggressive disease”, associated to better OS.
- Can the authors give a reference when they argue “browning of white adipose tissue makes poor prognosis of cancer cachexia”? (line 65)
2) On methods
For statistics, the authors must confirm that they used non-parametric test since they have n=5 and n=10 mice.
3) Results
- The authors cannot use mean+/- SEM with n=5 or n=10 but median +/- SD.
- For all figures (1,2,3,4,5 and 6), they cannot use histogram, they must use scatter plot on individual value with mean +/- SD (as in figure 1C but with mean+/-SD).
- For figures 2B and 4E, the authors must quantified adipocytes size and number.
- Did the mice from the group with severe cachexia (carcinomatosis) have ascitic fluid?
- There is absolutely no interest in comparing body size on image. Please remove from line 246 to 248
- This is no significant difference in survival curve between groups in figure 3C, so the authors cannot indicate that mortality of WAF and EtAF groups was improved.
- The authors should be carefully on word used. Line 264 they stated that “WAF administration suppressed the decrease of gMuscle size (no quantification so useless) and weight”. Firstly, authors must remove Figure 4a and secondly use appropriate word for the effect of WAF on muscle weight. There is no suppression of the muscle loss.
- Concerning in vitro experiment, surprisingly the authors did not indicate the number of experiment performed for each parameters. It is crucial for statistical analyses.
- Figure 5 E: the authors must add a graphic and statistics for MAFbx, Murf-1 and MyH protein expression in Control/CM/ and WAF/etaF-CM cells. Are blots saturated for MAFBx?
- Line 299 Figure 5 title. The authors did not measured C2C12 degradation. Please change the title.
- Figure 6A and B, the authors used oil red O solution to quantify differentiation of 3T3-L1 cell. The authors should use a more specific marker of adipocyte differentiation.
4) Conclusion
The authors should emphasized that these data are from mice: “AF prevented wasting muscle and adipose tissue by suppressing systemic inflammation” in mice. There is large differences between mice and patients. Before making plan to give tea to cancer patients in order to fight cachexia, the authors should give information on doses used in mice, on cells in comparison to what can be expected for patients. For a patient of 70 kg (using 100mg/kg/day) that means 7 grams of WAf or EtAF per day. How many kilograms of AF will be needed?
Author Response
Reviewer 2.
1) In the introduction
- the authors had forgotten that skeletal muscle loss is part of the parameters that reduce efficacy of anti-cancer treatments (Sjoblom B Lung Cancer 2015, Degens 2019) line 43.
→ As your comment, the references were added. [5] Sjoblom, B.; Gronberg, B.H.; Benth, J.S.; Baracos, V.E.; Flotten, O.; Hjermstad, M.J.; Aass, N.; Jordhoy, M. Low muscle mass is associated with chemotherapy-induced haematological toxicity in advanced non-small cell lung cancer. Lung Cancer 2015, 90, 85-91, doi:10.1016/j.lungcan.2015.07.001. [6] Degens, J.; Sanders, K.J.C.; de Jong, E.E.C.; Groen, H.J.M.; Smit, E.F.; Aerts, J.G.; Schols, A.; Dingemans, A.C. The prognostic value of early onset, CT derived loss of muscle and adipose tissue during chemotherapy in metastatic non-small cell lung cancer. Lung Cancer 2019, 133, 130-135, doi:10.1016/j.lungcan.2019.05.021.
- The authors used the reference of Gullett et al. to indicate that “approximately 20% of them died directly because of cachexia” (lines 46-47). It is interesting since Gullett et al cited Inui et al 2002 and MacDonald et al. 2003. Inui and his collaborators cited Bruera E. 1997. With all the respect that I have for Eduardo Bruera, this review did not provide any data on mortality related to cancer cachexia. Concerning Mc Donalds’ article I did not find data on mortality too. The authors need to find the adequate reference if they want to argue about direct effect of cachexia on patients’ mortality. Be careful, please read the articles!
→ As your comment, we cited the adequate reference.
[7] Blum, D.; Stene, G.B.; Solheim, T.S.; Fayers, P.; Hjermstad, M.J.; Baracos, V.E.; Fearon, K.; Strasser, F.; Kaasa, S.; Euro-Impact Collaborators. Validation of the Consensus-Definition for Cancer Cachexia and evaluation of a classification model--a study based on data from an international multicentre project (EPCRC-CSA). Ann Oncol 2014, 25, 1635-1642, doi: 10.1093/annonc/mdu086.
- the authors claim that in cancer patient MuRF1 and MAFbx are significantly increased and then can be considered as markers of muscle atrophy ref [7] line 51-52. Again, the authors should have read the review of Bodine SC, since data reported on Mafbx expression in muscle during cancer cachexia extracted from preclinical model. Again, the authors must use the appropriate reference by looking carefully to the literature, since there is no really consensus on MuRF1 and MAFbx increase in cancer patient skeletal muscle.
→ According to Yuan et al. 2015, Atrogin-1 and MuRF1 in rectus abdominis muscle of the malignant disease patients were increased compared to the benign disease patients, respectively. This reference was added.
[9] Yuan, L.; Han, J.; Meng, Q.; Xi, Q.; Zhuang, Q.; Jiang, Y.; Han, Y.; Zhang, B.; Fang, J.; Wu G. Muscle-specific E3 ubiquitin ligases are involved in muscle atrophy of cancer cachexia: an in vitro and in vivo study. Oncol. Rep. 2015, 33, 2261-2268, doi: 10.3892/or.2015.3845.
- Line 58, the authors cite Shachar S. Again the authors need to read very carefully this article. In fact (see Table 2) on 4 clinical studies 2 concluded that high SAT is associated with worse survival, and one (Antoun S et al.) concluded that “It seems that a high volume of SAT could be a metabolic hallmark of less aggressive disease”, associated to better OS.
→ As your comment, we cited the adequate reference.
[12] Antoun, S.; Bayar, A.; Ileana, E.; Laplanche, A.; Fizazi, K.; di Palma, M.; Escudier, B.; Albiges, L.; Massard, C.; Loriot, Y. High subcutaneous adipose tissue predicts the prognosis in metastatic castration-resistant prostate cancer patients in post chemotherapy setting. Eur. J. Cancer. 2015, 51, 2570-2577. doi: 10.1016/j.ejca.2015.07.042.
- Can the authors give a reference when they argue “browning of white adipose tissue makes poor prognosis of cancer cachexia”? (line 65)
→ We cited this sentence ‘browning of white adipose tissue makes poor prognosis of cancer cachexia’ using mice results. But prognosis mean human results. Because of this, we also think it made a misunderstanding. So, we edited the sentence and added references. We are sorry for making misunderstanding.
[13] Petruzzelli, M.; Schweiger, M.; Schreiber, R.; Campos-Olivas, R.; Tsoli, M.; Allen, J.; Swarbrick, M.; Rose-John, S.; Rincon, M.; Robertson, G., et al. A switch from white to brown fat increases energy expenditure in cancer-associated cachexia. Cell Metab 2014, 20, 433-447, doi:10.1016/j.cmet.2014.06.011. [17] Kir, S.; Spiegelman, B.M. Cachexia & Brown Fat: A Burning Issue in Cancer. Trends Cancer 2016, 2, 461-463, doi:10.1016/j.trecan.2016.07.005.
2) On methods
For statistics, the authors must confirm that they used non-parametric test since they have n=5 and n=10 mice.
→ As you commented, we used Kruskal–Wallis test with Dunn’s multiple comparison as a nonparametric test. We mentioned in Material & Methods section and every figure legends.
3) Results
- The authors cannot use mean+/- SEM with n=5 or n=10 but median +/- SD.
- For all figures (1,2,3,4,5 and 6), they cannot use histogram, they must use scatter plot on individual value with mean +/- SD (as in figure 1C but with mean+/-SD).
→ As you commented, we used scatter plot on individual value with mean +/- SD in in vivo results. Since we also misused SEM and SD in several figures, we edited all the figures using Mean with SD. But, figures 5 and 6 are almost PCR in vitro data. Most researcher commonly used histogram for their in vitro data. If graph changes are still needed more for in vitro results, please let us know your opinion.
- For figures 2B and 4E, the authors must quantified adipocytes size and number.
→ We quantified adipocytes size using Image J. Also, we counted The number of cells per field as other reserachers (Yadhu Sharma; 2019, mol thera). The number of cells per field was significantly increased in tumor mice, confirming an increased adipocyte size when compared to the control group. We added in figure 2F,G (Line 239-244) and figure 5C,D (Line 300-302)
- Did the mice from the group with severe cachexia (carcinomatosis) have ascitic fluid?
→ Although we did not record ascitic fluid levels, ascitic fluid was observed when sacrifice the mice.
- There is absolutely no interest in comparing body size on image. Please remove from line 246 to 248
→ As you commented, we removed the image.
- This is no significant difference in survival curve between groups in figure 3C, so the authors cannot indicate that mortality of WAF and EtAF groups was improved.
→ As you commented, we changed the sentence using ‘decreased’, not the ‘improved’. Also, we indicated the necessary of extra experiment for significant difference.
- The authors should be carefully on word used. Line 264 they stated that “WAF administration suppressed the decrease of gMuscle size (no quantification so useless) and weight”. Firstly, authors must remove Figure 4a and secondly use appropriate word for the effect of WAF on muscle weight. There is no suppression of the muscle loss.
→ As you commented, we edited the result. We did statistical test again using nonparametric methods. From there, there was no significant different in muscle data.
- Concerning in vitro experiment, surprisingly the authors did not indicate the number of experiment performed for each parameters. It is crucial for statistical analyses.
→ Results are from 3 independent experiments. We added this comment in Figure legends 6, 7, and 8.
- Figure 5 E: the authors must add a graphic and statistics for MAFbx, Murf-1 and MyH protein expression in Control/CM/ and WAF/etaF-CM cells. Are blots saturated for MAFBx?
→ As you commented, we added density graph for MAFbx, Murf-1 and MyH protein expression. We did not used any saturated blot image, but we changed the western blot images for avoiding any misunderstanding (Figure 7).
- Line 299 Figure 5 title. The authors did not measured C2C12 degradation. Please change the title.
→ As you commented, we edited Figure 5 title (Current Figure 6 title).
- Figure 6A and B, the authors used oil red O solution to quantify differentiation of 3T3-L1 cell. The authors should use a more specific marker of adipocyte differentiation.
→ Adipocyte differentiation is affected by specific genes and proteins related to adipogenesis. During the differentiation of 3T3-L1 cells, C/EBPa and PPARg play key roles as major transcription factors. The expression of C/EPBa cross-regulates with PPARg through a positive feedback loop, and transactivation downstream of several target genes, such as fatty acid synthesis (FAS), lipoprotein lipase (LPL), and adipocyte lipid binding protein (aP2), which are adipocyte specific and involved in maintaining the adipocyte phenotype. These factors are commonly used for determining whether adipocyte differentiated or not. We checked these factors using PCR and western blot (Figure 8) and described in Discussion section (Line 418-422).
4) Conclusion
The authors should emphasized that these data are from mice: “AF prevented wasting muscle and adipose tissue by suppressing systemic inflammation” in mice. There is large differences between mice and patients. Before making plan to give tea to cancer patients in order to fight cachexia, the authors should give information on doses used in mice, on cells in comparison to what can be expected for patients. For a patient of 70 kg (using 100mg/kg/day) that means 7 grams of WAf or EtAF per day. How many kilograms of AF will be needed?
→ For the 7 grams of WAf or EtAF per day, 100 g and 50 g of AF is required for clinical administration of WAF and EtAF, respectively. We also conducted serum analysis for confirming hepatotoxicity and nephrotoxicity. The mice treated with WAF and EtAF improved hepatotoxicity (AST, ALT) and nephrotoxicity (creatinine, BUN) induced by cancer cachexia (Table 2). We added these contents in discussion (Line 434-436) and conclusion section (line 468-472).

Round 2
Reviewer 1 Report
The authors have responded adequately to all the questions that I have recommended and by agreeing with their answers the article has, in my opinion, reached the possibility of being accepted for publication in the journal Nutrients, although for this, the authors should respond and make the two proposals that I make:
1.- Section 3.8 HPLC analysis, this is not the appropriate place since it reveals the bioactive compounds of the extract that cause the effects that this work tries to demonstrate and therefore should transfer this section to the beginning of the results (3.1) o , and I think it would be much more appropriate in the material and methods section, concretely after section 2.2. Extract of AF, becoming section 2.3.
2.- Because, according to the authors, those responsible for the effects they study are mainly due to the compounds, arctiin and arctigenin, the authors, therefore, should incorporate the corresponding structures of these compounds into the article. Let us remember the extraordinary relationship that exists between the structure of a compound and its biological function.
Author Response
1. Section 3.8 HPLC analysis, this is not the appropriate place since it reveals the bioactive compounds of the extract that cause the effects that this work tries to demonstrate and therefore should transfer this section to the beginning of the results (3.1), and I think it would be much more appropriate in the material and methods section, concretely after section 2.2. Extract of AF, becoming section 2.3.
→ We transferred the section 3.8 HPLC analysis to section 3.1. And the 2.13. HPLC Analysis of AF in the material and methods section was transferred to section 2.3.
2. Because, according to the authors, those responsible for the effects they study are mainly due to the compounds, arctiin and arctigenin, the authors, therefore, should incorporate the corresponding structures of these compounds into the article. Let us remember the extraordinary relationship that exists between the structure of a compound and its biological function.
→ As your comment, the structures and biological functions of the arctiin and arctigenin were described in Line 205-209.
